# Salivary Stress Biomarkers (Chromogranin A and Secretory IgA): Associations with Anxiety and Depressive Symptoms in Healthcare Professionals

**DOI:** 10.3390/nursrep16010003

**Published:** 2025-12-23

**Authors:** Tanya Deneva, Youri Ianakiev, Snezhana Stoencheva

**Affiliations:** 1Department of Clinical Laboratory, Medical University of Plovdiv, University Hospital “St. George”, 4002 Plovdiv, Bulgaria; snezhana.stoencheva@mu-plovdiv.bg; 2Research Institute at Medical University of Plovdiv, 4002 Plovdiv, Bulgaria; 3Department of Psychology, University of Plovdiv “Paisii Hilendarski”, 4000 Plovdiv, Bulgaria; ianakiev@uni-plovdiv.bg

**Keywords:** salivary biomarkers, chromogranin A, secretory immunoglobulin A, healthcare professionals, shift work, occupational stress, anxiety, depression

## Abstract

**Background/Objectives**: Shift-working healthcare professionals are exposed to high psychophysiological demands associated with occupational stress, anxiety, and depressive symptoms. Salivary chromogranin A (sCgA) and secretory immunoglobulin A (sIgA) are non-invasive biomarkers reflecting sympathetic nervous system activation and mucosal immune function, respectively, and are increasingly used to assess biological stress responses. This study examined changes in these biomarkers and their associations with anxiety and depression. **Methods**: This cross-sectional comparative observational study was conducted among healthcare professionals working 12-h shifts (n = 95) and non-shift-working controls (n = 95) and included a within-shift pre-post assessment, with saliva samples collected before and after the work shift. Salivary biomarkers were determined using ELISA methods. Anxiety and depression were assessed using the State–Trait Anxiety Inventory and the Zung Self-Rating Depression Scale. Data were analyzed with *t*-tests, correlation, and multiple linear regression. Statistical analyses included between- and within-group comparisons, correlation analyses, and multiple linear regression models to examine independent associations between salivary biomarkers and psychological outcomes. **Results**: After a 12-h shift, healthcare professionals showed increased sCgA (3.82 ± 0.95 vs. 4.68 ± 1.02 ng/mL; *p* < 0.001) and decreased sIgA (165.3 ± 32.4 vs. 142.6 ± 29.8 mg/dL; *p* < 0.001). Psychological scores were higher in healthcare professionals than in controls (*p* < 0.001). Salivary sCgA correlated positively with anxiety and depression (r = 0.41 to 0.45), while sIgA correlated negatively (r = −0.29 to −0.36). Regression analysis confirmed occupational group (healthcare professionals vs. controls) as the strongest predictor, with independent contributions of sCgA and sIgA to psychological scores. **Conclusions**: A 12-h work shift in healthcare professionals leads to increased salivary chromogranin A, indicating sympathetic activation, and decreased secretory IgA, reflecting reduced mucosal immune activity. The combined assessment of sCgA and sIgA provides a sensitive and non-invasive approach for monitoring occupational stress and identifying early risks of anxiety and depressive symptoms among shift-working healthcare professionals.

## 1. Introduction

Mental stress, anxiety, and depressive symptoms are common among healthcare professionals, particularly under conditions of high workload, extended shifts, and emotional exhaustion. The professional environment of healthcare workers is characterized by high intensity, organizational pressure, and the constant need to make critical decisions. These factors, combined with night duties and prolonged working hours, can lead to substantial psychophysiological stress and increase the risk of anxiety, depression, and burnout. Pandemic-related crises have further intensified these demands and demonstrated that salivary biomarkers can capture dynamic changes in the stress response among both hospitalized patients and healthcare staff [1,2,3].

Systematic reviews and meta-analyses have highlighted that secretory immune markers and neuroendocrine mediators in saliva are sensitive to both acute and chronic stress, following distinct temporal patterns [2,4,5,6]. Within the biological framework of stress, the hypothalamic–pituitary–adrenal (HPA) axis and the sympathoadrenal system play central roles. While salivary cortisol reflects HPA axis activity, salivary chromogranin A (sCgA) is co-released with catecholamines from chromaffin granules and serves as an indicator of sympathetic activation, whereas secretory immunoglobulin A (sIgA) is a key effector of mucosal immunity sensitive to psycho-emotional strain [4,5,6,7,8,9].

Recent comparative studies have shown that sCgA complements the information provided by cortisol and α-amylase, serving as an additional marker of autonomic reactivity to mental and occupational stress, including in hospital settings [7,8,9,10]. Early experimental data demonstrated an increase in sCgA during cognitively induced stress [8], while more recent studies on occupational fatigue and shift work have described associations between sCgA levels and subjective fatigue [9,10]. Additional findings suggest that morning sCgA levels correlate with perceived fatigue and chronic complaints [11].

Secretory IgA reflects the state of mucosal defense and responds sensitively to fluctuations in psycho-emotional balance. During acute stress, sIgA typically increases transiently, whereas prolonged psychosocial distress leads to decreased levels [4,5,6,12]. Experimental and observational studies indicate that sIgA dynamics depend on the duration and context of stress, as well as on immunoglobulin subclasses and the secretory component [5,13]. Within the field of psychoneuroimmunology, the convergence of findings on sIgA and inflammatory markers supports the use of salivary biomarker panels in stress research [6].

Shift work among healthcare professionals is associated with circadian disruption, sleep disturbances, fatigue, and heightened anxiety—manifestations of the so-called “shift work disorder” [1,2]. Studies in hospital staff and other occupational groups have shown that night shifts and extended work hours are related to adverse physiological changes, a higher likelihood of errors, and an increased risk of burnout [1,2,10].

Research investigating anxiety among healthcare workers during crisis situations, including the COVID-19 pandemic, has reported elevated levels of state and trait anxiety, associated with employment type and work schedule [14,15,16,17]. These findings are consistent with broader evidence on stress-related salivary biomarkers in shift workers, including cortisol, DHEA-S, and immune parameters [4,13,18,19].

Psychological assessment in such studies relies on validated instruments. The Zung Self-Rating Depression Scale (ZSDS) is widely used in clinical and population studies, with recent analyses refining its validity and screening thresholds [16,20,21,22]. The State–Trait Anxiety Inventory (STAI) is a standardized tool with robust psychometric properties, suitable for assessing anxiety among healthcare professionals [5,14,15,16,17,23]. Combining subjective self-report scales with objective salivary biomarkers provides an integrated perspective on the psychophysiological response to stress.

The available literature on sCgA and sIgA in healthcare professionals shows considerable heterogeneity, largely due to differences in study protocols, preanalytical factors (e.g., time of sampling, food and caffeine intake, hydration), and methodological variations. Despite this variability, consistent patterns have been reported: sCgA tends to increase in response to acute psychological and occupational stress, whereas sIgA appears to be more sensitive to prolonged or chronic stress exposure and generally decreases over time, although transient increases during acute stressors have also been described [4,5,6,8,9,10,11,12,24,25]. In this context, the growing interest in multimarker salivary stress panels (e.g., cortisol, α-amylase, sCgA, sIgA) highlights the need for integrative approaches that combine biological markers of stress with standardized psychometric assessments [2,16,19]. However, most existing studies have examined these biomarkers independently. The present study fills this gap by jointly examining changes in the salivary stress biomarkers sCgA and sIgA, along with standardized psychometric assessments of anxiety and depressive symptoms in healthcare professionals working under increased workload, thereby contributing to a more comprehensive and integrated understanding of the dynamic interaction between biological stress responses and psychoemotional outcomes under conditions of acute work-related stress.

Therefore, the aim of this study was to examine the changes in sCgA and sIgA levels in saliva before and after a 12-h work shift in healthcare professionals, compared with a control group, and to explore their relationships with symptoms of anxiety and depression. We hypothesized that healthcare professionals would show increased sCgA and decreased sIgA levels after the work shift, accompanied by higher anxiety and depression scores.

## 2. Materials and Methods

### 2.1. Study Design and Participants

This cross-sectional comparative observational study was conducted between August 2022 and December 2023 at the University Hospital “St. George”, Plovdiv, Bulgaria, and included a within-shift pre–post assessment before and after a 12-h work shift.

The study included two parallel groups:Healthcare professionals’ group (n = 95)—physicians, nurses, and medical laboratory technicians employed at the University Hospital “St. George”, which operates under an emergency status with continuous 24/7 service and a rotational 12-h shift schedule.Control group (n = 95)—healthy volunteers recruited during routine laboratory examinations who met the inclusion criteria (full-time employment, no night or shift work, and no significant occupational stress exposure).

Inclusion criteria for healthcare professionals were employment as a physician, nurse, or medical laboratory technician; active engagement in rotational 12-h shift work for at least one year prior to participation; full-time employment status; age between 25 and 60 years; absence of chronic somatic or psychiatric illness; and no current use of medications known to affect neuroendocrine or immune function.

Inclusion criteria for the control group were full-time employment in non-healthcare occupations; no history of night or shift work within the preceding 12 months; absence of chronic illness or regular medication affecting salivary biomarkers; no reported exposure to significant occupational stress; and provision of written informed consent.

Participants were recruited through voluntary invitation among hospital staff and routine laboratory visitors. Of the 210 individuals approached, 190 met the inclusion criteria and agreed to participate (response rate: 90.5%).

The sample size was estimated a priori using G*Power 3.1 software to detect a medium effect size (Cohen’s d = 0.5) for between-group differences in psychological outcomes, with α = 0.05 and statistical power (1 − β) = 0.80. A minimum of 85 participants per group was required; to ensure adequate power and compensate for possible dropouts, 95 participants were included in each group. The choice of a medium effect size was based on Cohen’s conventional criteria and reflects common methodological practice in psychophysiological and occupational stress research when prior population-specific data are limited. In the absence of population-specific preliminary data, this approach is considered appropriate and conservative. Importantly, the observed effect sizes in the present study were equal to or exceeded the assumed effect size, supporting the adequacy of the sample size.

The study protocol was approved by the Scientific Ethics Committee of “Paisii Hilendarski” University of Plovdiv (Approval No. RD-38-134/08/06/2022) and was conducted in accordance with the principles of the Declaration of Helsinki.

### 2.2. Saliva Sample and Analysis of Salivary Biomarkers

Saliva samples were collected twice from each participant—at the beginning of the working day (between 06:00 and 08:00), before the start of the 12-h shift, and immediately after its completion. In the control group, samples were collected at equivalent times under standard daytime conditions to ensure comparability of the circadian biomarker profiles.

Sample collection was performed using Salivette^®^ systems (Sarstedt, Nümbrecht, Germany), in accordance with the standard preanalytical requirements. Participants were instructed to refrain from food, drinks, and caffeine intake for at least 30 min before sampling. Samples were stored at −20 °C until analysis, but for no longer than two months, following the manufacturer’s recommendations.

### 2.3. Psychological Assessment

The psychological profile of participants was evaluated using three standardized and validated self-report instruments. Depressive symptoms were assessed with the Zung Self-Rating Depression Scale (ZSDS), widely used in clinical and epidemiological studies for screening and quantitative evaluation of depression.

Anxiety was measured using the Spielberger State–Trait Anxiety Inventory (STAI), which consists of two subscales: STAI-State (state anxiety) and STAI-Trait (trait anxiety), reflecting both momentary emotional state and stable predisposition to anxiety [14,16,17,20,21,22].

Depressive symptoms and anxiety were assessed using Bulgarian-language versions of the respective standardized self-report instruments, applied according to the original structure, scoring procedures, and interpretation guidelines. Both instruments have well-established psychometric properties in clinical and occupational research. The ZSDS demonstrates good internal consistency (Cronbach’s α = 0.84–0.92) and good test–retest reliability, while the STAI shows excellent reliability (α = 0.88–0.94) and strong construct validity across diverse populations.

According to standard interpretation criteria, ZSDS total scores below 50 indicate absence of depressive symptoms, 50–59 correspond to mild depression, and 60–69 to moderate depression [21,22]. For the STAI-State and STAI-Trait scales, scores below 40 indicate no anxiety, 40–55 moderate anxiety, and above 55 high anxiety [17].

The STAI and ZSDS questionnaires were self-administered by participants in a quiet setting at the workplace. Standardized instructions were provided, and a trained researcher was present to clarify procedural questions without influencing responses.

### 2.4. Statistical Analysis

All data were analyzed using SPSS software (version 27, IBM Corp., Armonk, NY, USA) and R software (version 4.4.1, R Foundation for Statistical Computing, Vienna, Austria). The threshold for statistical significance was set at *p* < 0.05 (two-tailed).

Descriptive statistics were calculated to summarize the characteristics of the study sample. Continuous variables were expressed as mean ± standard deviation (SD) or as median with interquartile range (IQR), depending on data distribution, while categorical variables were presented as frequencies and percentages.

Effect sizes were calculated using Cohen’s d and interpreted according to conventional criteria, with values of approximately 0.2 indicating small effects, 0.5 medium effects, and ≥0.8 large effects.

Before performing inferential analyses, normality of distribution was examined using the Shapiro–Wilk test, and homogeneity of variances was checked using the Levene test. sCgA and STAI scores were approximately normally distributed, whereas sIgA and ZSDS total scores deviated from normality and were analyzed using non-parametric tests where appropriate.

To assess changes within groups (before- and after-shift), paired *t*-tests were used for normally distributed data, whereas the Wilcoxon signed-rank test was applied when normality assumptions were not met. Between-group differences were examined using Student’s *t*-test or Welch’s *t*-test in cases of unequal variances; for non-normally distributed data, the Mann-Whitney U test was employed. For within-group pre-post comparisons, Cohen’s d was reported to allow comparability with between-group effect sizes and with previous literature, although paired-sample effect size measures may also be considered.

Baseline comparability between groups was examined prior to outcome analyses, and no statistically significant baseline differences were observed. Although analysis of covariance (ANCOVA) represents a robust method for adjusting baseline values, the analytical approach was complemented by multivariable linear regression models including relevant covariates (age, sex, and years of work experience), thereby accounting for potential baseline-related confounding effects.

The relationships between psychometric and biochemical indicators were explored through correlation analyses. Pearson’s correlation coefficient was used for normally distributed variables, and Spearman’s rank correlation for those not meeting normality assumptions.

To identify independent predictors of psychological outcomes, multiple linear regression analyses (OLS) were conducted, with ZSDS total scores, STAI-State, and STAI-Trait serving as dependent variables. Predictor variables included sCgA and sIgA after work shift, group (healthcare professionals vs. controls), age, gender, and years of work experience. Before conducting multiple linear regression analyses, the assumptions of linearity, normality of residuals, homoscedasticity, and absence of multicollinearity were assessed. Linearity and homoscedasticity were assessed by inspecting the residual plots, while normality of residuals was examined using Q-Q plots. Multicollinearity was assessed using variance inflation factors (VIF), with all values remaining below generally accepted thresholds. No significant violations of regression assumptions were found.

To account for multiple testing, the Benjamini-Hochberg false discovery rate (FDR) control procedure was applied to correlation and multiple linear regression analyses examining associations between salivary biomarkers and psychological measures. The FDR approach was preferred over the Bonferroni correction to avoid excessive loss of statistical power when analyzing multiple, partially correlated biological and psychometric variables.

Incomplete questionnaires or saliva samples with missing data (<5%) were excluded from the analysis; no data imputation was performed.

## 3. Results

The study included a total of 190 participants, evenly divided between healthcare professionals (n = 95) and controls (n = 95). The two groups did not differ significantly in terms of sex (*p* = 0.396), age (*p* = 0.394), or years of professional experience (*p* = 0.362). The mean age of participants was 45.6 years (SD = 4.6), and the mean duration of work experience was 20.4 years (SD = 5.0). These comparable demographic profiles allowed for the interpretation of salivary biomarkers and psychological findings without substantial confounding effects from age, sex, or work experience.

Descriptive statistics, between-group differences, and effect sizes (Cohen’s d) for all biochemical and psychometric parameters are presented in Table 1.

The results demonstrated statistically significant between-group differences in both biochemical and psychometric indicators. Among healthcare professionals, a pronounced increase in sCgA was observed following the 12-h work shift (*p* < 0.001, d = 0.91) as well as, conversely, significantly decreased sIgA levels (*p* < 0.001, d = 0.75).

Psychometric assessments (ZSDS total scores, STAI-State, STAI-Trait) also revealed statistically significant between-group differences, with healthcare professionals exhibiting markedly higher anxiety and depression scores (*p* < 0.001, d > 1.5).

To further evaluate the within-group physiological dynamics, paired analyses were performed for healthcare professionals before and after their 12-h shifts. Post-shift measurements revealed a statistically significant increase in sCgA (*p* < 0.001, d = 0.73) and a decrease in sIgA (*p* < 0.001, d = 0.66) (Table 2). Both effects were of moderate to high magnitude, indicating a pronounced acute physiological stress response resulting from prolonged duty hours.

Correlation analyses were conducted to assess the relationships between salivary biomarkers and psychological states. Table 3 presents the linear relationships between salivary biomarkers and psychometric indicators in healthcare professionals. After the 12-h shift, a significant positive correlation was found between sCgA concentrations and scores on anxiety and depression scales (r = 0.41 to 0.45, *p* < 0.01). Conversely, sIgA showed a negative correlation with the same psychometric indicators (r = −0.29 to −0.36, *p* < 0.05). The strongest significant relationship was a positive correlation between post-shift sCgA levels and state anxiety measured by the STAI-State (r = 0.45, *p* < 0.01) (Figure 1).

To assess the influence of psychological and biomarker factors on anxiety and depression levels, a multiple linear regression analysis (OLS) was conducted. The psychometric scales (STAI-state, STAI-trait, ZSDS total scores) and salivary biomarkers (sCgA, sIgA) were included in the model to determine their predictive effect on emotional states.

Three multiple linear regression models were calculated to assess the effect of group affiliation (health professionals vs. control group) on psychological indicators, controlling for age, gender, and work experience (Table 4)

The depression model (ZSDS total scores) showed high explanatory power (R^2^ = 0.72; Adj. R^2^ = 0.71), with group affiliation being a significant predictor. The situational anxiety model (STAI-state) was also statistically significant, explaining 64% of the variance in the results (R^2^ = 0.64; Adj. R^2^ = 0.63). The personality anxiety model (STAI-trait) had the highest predictive value, explaining 79% of the variance (R^2^ = 0.79; Adj. R^2^ = 0.78).

Multiple linear regression analyses (OLS) showed that group affiliation (health professionals vs. controls) was the strongest predictor across all models (*p* < 0.001). After controlling for demographic variables (age, gender, work experience), the effects of the biomarkers sCgA_after and sIgA_after decreased but remained statistically significant, albeit with low power, confirming their independent contribution as physiological predictors of anxiety and depression.

In the depression model (ZSDS total scores), higher levels of sCgA_after (β = 0.15, *p* = 0.035) and lower levels of sIgA_after (β = −0.06, *p* = 0.017) were associated with increased depressive symptoms.

In the state anxiety model (STAI-state), sCgA_after remained significant (*p* = 0.048), indicating a link between momentary anxiety and neuroendocrine reactivity.

In the trait anxiety model (STAI-trait), sIgA_after emerged as a negative predictor (*p* = 0.026), suggesting that lower immune activity was related to higher chronic anxiety.

Although these effects were modest, they were consistent across models and occurred in parallel with a clear group difference between healthcare professionals and controls.

## 4. Discussion

In the present study, significant changes were observed in salivary stress biomarkers and psychometric indicators among healthcare professionals compared with the control group. The increase in salivary chromogranin A levels after the 12-h work shift suggests activation of the sympathoadrenal system and an intensified physiological stress response. At the same time, the decrease in secretory immunoglobulin A may indicate a stress-related immunosuppressive effect due to prolonged exposure to occupational strain.

These findings align with previous research showing that sCgA rises under sustained work pressure and sleep deprivation [27,28], while sIgA tends to decline with chronic fatigue and stress [10]. Similar associations have been described by Black et.al. [29], linking occupational stress among healthcare workers with endocrine and immune dysregulation.

Overall, healthcare professionals showed moderate anxiety (both state and trait) and a tendency toward mild depressive symptoms, whereas the control group maintained scores within normal ranges. The large effect sizes (d > 1.5) highlight the robustness of these differences and reinforce the assumption that occupational stress has a substantial impact on the psychological well-being of medical personnel. These results are consistent with prior studies [27,29] that have identified similar psychoemotional patterns in healthcare workers exposed to prolonged stress.

Within the framework of McEwen’s psychophysiological model of stress [30], the observed alterations likely reflect an adaptive emotional and cognitive response to chronic occupational stressors associated with extended wakefulness, high responsibility, and circadian rhythm disruption.

This dynamic confirms the activation of the sympathoadrenal axis and the transient suppression of mucosal immunity in healthcare professionals exposed to prolonged occupational stress. The analysis of salivary biomarkers demonstrated a clear increase in chromogranin A and a decrease in secretory immunoglobulin A levels after a 12-h work shift, whereas no significant changes were observed among the control participants. These findings reflect an acute physiological stress response induced by the workload and prolonged professional exposure characteristic of healthcare occupations.

The results of the present study are consistent with previous reports on the physiological response to acute stress, in which sympathetic activation is typically accompanied by a suppression of immune secretion. Similar patterns of increased sCgA and decreased sIgA concentrations among healthcare workers engaged in night or extended shift work have been reported by Abeywickrama et al. [27], Rohleder et al. [28], and Takatsuji et al. [31].

The strongest and most statistically significant association between post-shift sCgA levels and state anxiety (STAI-State; r = 0.45, *p* < 0.01) confirms that physiological sympathoadrenal activation is closely linked to the subjective experience of anxiety and psychoemotional stress. This relationship reflects the classical model of psychoneuroimmunological regulation, in which activation of the sympathetic nervous system and the adrenomedullary axis results in increased secretion of sCgA accompanied by a concomitant suppression of the mucosal immune response, as indicated by decreased sIgA levels [28,32]. Morita et al. [10] reported that individuals engaged in long-term night work exhibited elevated sCgA and reduced sIgA concentrations, along with poorer sleep quality and increased perceived stress. The authors interpreted these findings as evidence of chronic sympathetic activation combined with circadian disruption, leading to sustained suppression of immune function. In addition to its well-documented association with chronic stress, the observed post-shift reduction in sIgA may also reflect a rapid, stress-induced modulation of mucosal immunity. Acute activation of the sympathoadrenal system is known to transiently suppress salivary immunoglobulin secretion through autonomic and neuroendocrine pathways, likely mediated by catecholamine release and altered salivary gland activity. From an adaptive perspective, this short-term suppression may represent a redistribution of physiological resources toward immediate coping and energy mobilization during intense work-related demands. Such mechanisms provide a plausible explanation for the rapid decline in sIgA observed after a single prolonged work shift.

These observations are consistent with the present results, which revealed a significant post-shift increase in sCgA and a negative correlation between sIgA and anxiety/depression scores after a 12-h shift. Rohleder [32] emphasized that during the transition from acute to subacute stress, an imbalance develops between sympathetic activation and immunoregulatory mechanisms, characterized by increased sCgA and reduced sIgA. This dual physiological pattern represents an adaptive response of the organism aimed at mobilizing energy resources while temporarily downregulating non-essential functions such as mucosal immunity. These mechanisms are supported by experimental and occupational studies demonstrating that mental and physical stressors are associated with increased sCgA levels and heightened subjective tension and anxiety [8,33,34,35].

In real-world settings of occupational stress, similar patterns have been reported in professional and academic settings. For example, Yamaguchi et al. [36] observed increased sCgA and decreased sIgA secretion in shift-working nurses, which was associated with fatigue and sleep disturbances. Consistent with these findings, the present data show that increased sCgA levels after prolonged shift work reflect activation of the sympathoadrenal axis and are closely related to subjectively experienced anxiety. The concomitant decrease in sIgA suggests stress-related immunosuppressive effects, highlighting the interplay between neuroendocrine and immune regulation. This psychoneuroendocrine-immune profile is consistent with established models of stress physiology and suggests that even short-term but intense occupational stress can induce physiological changes with potential health consequences. The results of the regression analysis provide deeper insight into the joint contribution of biological and psychological factors to anxiety and depression among healthcare professionals.

These findings expand upon the correlational results presented earlier by demonstrating that, even when controlling for demographic variables, biomarkers such as CgA_after and sIgA_after retained an independent, though weaker, predictive role.

Belonging to the healthcare professional group was confirmed as the main predictor of both anxiety and depression, emphasizing the role of occupational exposure and workload as key determinants of psycho-emotional health. Nevertheless, the biomarkers CgA_after and sIgA_after maintained independent associations with the psychological measures, indicating that stress possesses both biological and subjective dimensions.

The positive association between CgA_after and depression supports the role of chromogranin A as a marker of acute sympatho-adrenal activation under stress [37]. The negative relationship with sIgA_after is consistent with immune suppression during chronic psycho-emotional strain—an effect previously observed among healthcare professionals with high job strain [38]. This combination of enhanced neuroendocrine and reduced immune activity is characteristic of prolonged psychophysiological load and may represent a biological substrate of affective symptoms in occupational stress.

For situational anxiety, CgA_after remained a significant predictor, reflecting the acute physiological stress response, while sIgA_after did not reach significance. This pattern supports the interpretation that CgA reflects short-term sympathetic activation, linking momentary (state) anxiety with neuroendocrine reactivity [35] whereas sIgA is more involved in longer-term stress processes, consistent with previous findings showing correlations between sIgA, cortisol, chronic fatigue, and anxiety in nurses [2,36]

The dominant group effect suggests that exposure to the professional environment and demanding shifts contribute to elevated situational anxiety, regardless of biological variability.

In the trait anxiety (STAI_trait) model, sIgA_after was a significant negative predictor (*p* = 0.026), indicating a link between lower immune activity and higher chronic anxiety, as also reported by other authors [39]. This finding aligns with the allostatic load framework [30], in which chronic stress leads to lasting physiological dysregulation.

The findings of this study have direct relevance for mental health nursing practice and research. The demonstrated associations between salivary stress biomarkers and psychological indicators such as anxiety and depression underline the importance of early identification and prevention of work-related stress among healthcare professionals. Mental health nurses can play a key role in implementing regular psychophysiological screening and providing targeted support interventions to reduce emotional exhaustion and burnout. Incorporating non-invasive salivary biomarkers such as CgA and sIgA into occupational health assessments may enhance the effectiveness of stress management strategies. These results also highlight the need for further nursing research focused on resilience, coping, and the development of evidence-based programs promoting the well-being of shift-working healthcare staff.

## 5. Limitations and Strengths

Although the study yielded significant findings, several limitations should be acknowledged.

First, the cross-sectional design does not allow for causal inferences regarding the direction of the observed relationships between stress biomarkers and psychometric indicators. Longitudinal or experimental studies would be required to confirm the temporal dynamics of these associations.

Second, all participants were recruited from a single institutional setting, which may limit the generalizability of the findings to other healthcare institutions and professional settings.

Third, although salivary chromogranin A (CgA) and secretory immunoglobulin A (sIgA) are reliable non-invasive biomarkers of stress, their levels can be influenced by factors such as circadian rhythm, hydration, food intake, and individual differences in salivary flow, which were only partially controlled in this study. Circadian variability may have contributed to the variability in CgA and sIgA levels, along with the effects of work-related stress, and should be more strictly controlled in future studies through standardized sampling times and more stringent pre-collection protocols. In addition, relevant confounding factors such as sleep duration, pre-shift fatigue, chronotype, and lifestyle characteristics were not systematically assessed in this study. These factors may influence both psychophysiological responses and salivary biomarker levels, particularly in the context of prolonged or night work, and should be considered in future research.

Fourth, the psychological data were based on self-report instruments, which may be affected by social desirability or subjective response bias.

Although burnout-specific instruments were not included, anxiety and depressive symptoms represent key psychoemotional consequences of chronic occupational stress. Future studies may benefit from including measures of burnout in healthcare settings.

Finally, the study focused primarily on acute physiological changes following a single 12-h work shift; therefore, the results may not fully reflect the effects of chronic occupational stress or long-term adaptive mechanisms.

Despite the limitations, the findings remain significant. The combined psychobiological approach—integrating validated psychometric instruments with objective biochemical indicators—provides a more comprehensive and multidimensional assessment of stress among healthcare professionals. Our findings deliver reliable empirical evidence on the psychophysiological impact of prolonged work shifts and highlight the value of salivary biomarkers as non-invasive tools for evaluating occupational stress.

## 6. Conclusions

Healthcare professionals exhibited significantly higher levels of salivary chromogranin A (CgA) and lower levels of secretory immunoglobulin A (sIgA) after a 12-h shift, indicating physiological activation of the sympathoadrenal system and an immunosuppressive effect of prolonged stress exposure.

Psychometric results showed moderate anxiety and borderline depressive symptoms, corresponding to the biochemical alterations and reflecting the cumulative impact of occupational stress.

The observed between-group differences and significant correlations support the integrative psychoneuroimmunological model of stress. The regression models emphasize the predictive value of CgA and sIgA for psychological symptoms.

Taken together, the findings validate salivary biomarkers as sensitive, noninvasive indicators of psycho-emotional stress in healthcare professionals working under emergency and prolonged workload conditions.

Overall, the findings emphasize the importance of a multimarker approach to stress assessment that integrates psychological measures with biochemical indicators to capture both the subjective and physiological dimensions of stress.

## Figures and Tables

**Figure 1 nursrep-16-00003-f001:**
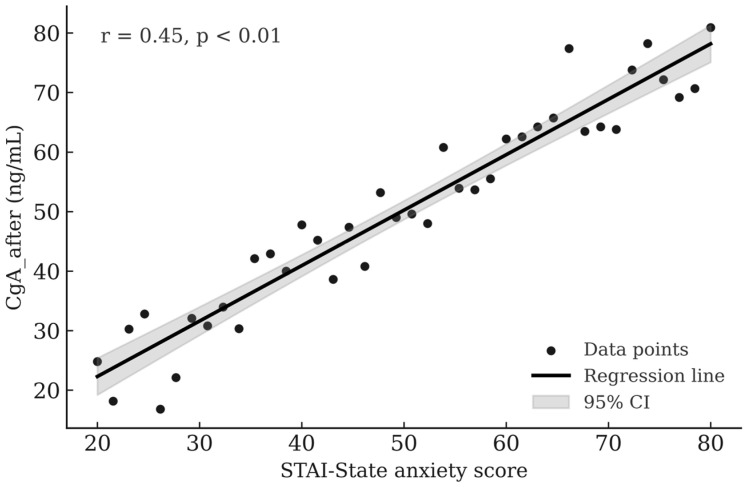
Correlation between sCgA_after and STAI-state anxiety scores in healthcare professionals.

**Table 1 nursrep-16-00003-t001:** Descriptive Statistics and Effect Sizes.

Parameter	Healthcare Professionals (n = 95) M ± SD	Control Group (n = 95) M ± SD	*p*-Value	Cohen’s d
CgA_before (ng/mL)	3.82 ± 0.95	3.76 ± 0.87	0.64	0.07
CgA_after (ng/mL)	4.68 ± 1.02	3.81 ± 0.91	<0.001	0.91
sIgA_before (mg/dL)	165.3 ± 32.4	163.1 ± 31.9	0.57	0.07
sIgA_after (mg/dL)	142.6 ± 29.8	164.5 ± 30.7	<0.001	0.75
ZSDS total	50.2 ± 7.5	36.4 ± 5.8	<0.001	1.74
STAI State	46.8 ± 8.3	34.7 ± 6.2	<0.001	1.64
STAI Trait	45.1 ± 7.9	33.9 ± 6.0	<0.001	1.57

Data are presented as mean values (M), standard deviations (SD), *p*-values from independent *t*-tests, and effect sizes (Cohen’s d) [26]. *p* < 0.05 was considered statistically significant. According to Cohen’s interpretation, d = 0.2 indicates a small effect, d = 0.5 a medium effect, and d ≥ 0.8 a large effect.

**Table 2 nursrep-16-00003-t002:** Within-group comparisons of salivary biomarkers in healthcare professionals (n = 95).

Biomarker	Before (Mean ± SD)	After (Mean ± SD)	t (94)	*p*-Value	Cohen’s d
sCgA (ng/mL)	3.82 ± 0.95	4.68 ± 1.02	−7.12	<0.001	0.73
sIgA (mg/dL)	165.3 ± 32.4	142.6 ± 29.8	6.48	<0.001	0.66

Data are presented as mean ± standard deviation (M ± SD). Within-group differences (before/after shift) were analyzed using the paired-samples *t*-test. *p*-values < 0.05 were considered statistically significant.

**Table 3 nursrep-16-00003-t003:** Correlations between biomarkers and psychometric scales in healthcare professionals (n = 95).

Biomarkers	ZSDS Total Scores	STAI-State	STAI-Trait
sCgA_before	0.22 (0.067)	0.19 (0.081)	0.17 (0.095)
sCgA_after	0.41 (0.002) **	0.45 (0.001) **	0.38 (0.004) **
sIgA_before	−0.18 (0.092)	−0.15 (0.116)	−0.20 (0.073)
sIgA_after	−0.36 (0.011) *	−0.31 (0.024) *	−0.29 (0.031) *

* *p* < 0.05; ** *p* < 0.01. The coefficients are presented as r (*p*), calculated using Pearson’s or Spearman’s correlation according to data normality.

**Table 4 nursrep-16-00003-t004:** Predictors of Anxiety and Depression.

Predictors	ZSDS Total Scores β (*p*)	STAI-State β (*p*)	STAI-Trait β (*p*)
CgA_after	0.145 (*p* = 0.035)	0.082 (*p* = 0.048)	0.056 (ns)
sIgA_after	−0.058 (*p* = 0.017)	−0.012 (ns)	−0.029 (*p* = 0.026)
Age	ns	ns	ns
Sex	ns	ns	ns
Years_experience	ns	ns	ns
Group (healthcare professionals vs. controls)	12.8 (*p* < 0.001)	8.94 (*p* < 0.001)	10.2 (*p* < 0.001)

Note: ns = not statistically significant (*p* ≥ 0.05). Models: ZSDS total scores R^2^ = 0.72 (Adj R^2^ = 0.71); STAI-state R^2^ = 0.64 (Adj R^2^ = 0.63); STAI-trait R^2^ = 0.79 (Adj R^2^ = 0.78). Models were calculated for the total sample (n = 190), including controls for age, gender, and work experience, and assessed the effect of group affiliation (health professionals vs. controls) on psychological and biological indicators.

## Data Availability

The datasets generated and analyzed during the current study are not publicly available due to ethical restrictions related to participant confidentiality. However, de-identified data may be made available from the corresponding author upon reasonable request and subject to approval by the appropriate institutional review board.

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
