# Peer review of "Salivary Stress Biomarkers (Chromogranin A and Secretory IgA): Associations with Anxiety and Depressive Symptoms in Healthcare Professionals"

_nursrep, 2025, doi:10.3390/nursrep16010003_

Round 1

Reviewer 1 Report

Comments and Suggestions for Authors

The article presented demonstrates a high relevance to establish the relationships between objective evaluations and self-report of health symptoms in work environments in the health area.

The introduction is clear, adequately addresses the most relevant conceptual aspects of the study, showing evidence regarding the purposes of the study. However, it is suggested that you justify why you concentrated and used general scales to measure depression and anxiety, instead of some measure of burnout, since you work in work contexts.

Methodologically, the design, instruments and procedures used are correct, showing an adequate sample size for the purposes of the study. 

There are no qualms regarding the presentation of the results. It is only suggested, to improve clarity, to incorporate graphs that allow the differences in the mean levels found in the variables in the previous and subsequent evaluations, as well as between the experimental and control groups (boxplot box graphs). 

The discussions and conclusions are supported by the results. It is noted that the authors recognize the limitations which generate projections for future studies.

Author Response

Dear Reviewer, 

Thank you for your positive and constructive evaluation of our manuscript.

Regarding the suggestion to justify the use of general measures of anxiety and depression rather than burnout-specific instruments, we would like to clarify that the main aim of the study was to investigate the relationships between salivary stress biomarkers and broadly defined psychoemotional symptoms that are comparable in professional and non-professional contexts. Anxiety and depressive symptoms represent major psychological outcomes of chronic occupational stress and are closely related to biological stress response mechanisms.

We agree that burnout is an important construct in occupational health research, especially in healthcare professionals. Accordingly, we have explicitly stated this point in the limitations section and note that future studies may benefit from including burnout-specific measures to further refine the assessment of work-related stress in healthcare professionals. The changes are highlighted in red (Section Limitation and strengths, line 470-472)

We also appreciate the suggestion to include graphical representations of the results. Given the current structure of the manuscript and the number of outcomes analyzed, the results were presented in a concise tabular format to ensure clarity and readability. However, we agree that graphical representations can further enhance the visualization of the data and will consider their inclusion in future work.

Reviewer 2 Report

Comments and Suggestions for Authors

Dear Authors,

Thank you for the opportunity to review the manuscript “Salivary Chromogranin A and Secretory IgA in Shift-Working Healthcare Professionals”, submitted to Nursing Reports. Below is my review, in which I evaluate the scientific quality, methodology, results, and relevance of the topic.

The manuscript presents an observational study that examines changes in chromogranin A (CgA) and secretory immunoglobulin A (sIgA) before and after a 12-hour shift among healthcare professionals, comparing them to non-shift workers. The topic is relevant, especially in the context of occupational health, and the study offers a significant contribution by integrating salivary biomarkers and psychological indicators.

The study is described as cross-sectional comparative observational [This cross-sectional comparative observational study... (p.3)]. However, data were collected before and after the shift, characterizing a pre-post assessment, which is closer to a quasi-experimental or short-term longitudinal observational study. It is recommended to review the nomenclature of the design or justify the classification as cross-sectional.

The authors acknowledge insufficient control of pre-analytical variables (circadian rhythm, hydration, food intake, and individual differences in salivary flow). It is recommended to further discuss the impact of these variables, especially since CgA and sIgA are sensitive to circadian variability.

The text is vague in its description of the randomization process for the controls. According to the authors, “healthy volunteers were randomly selected during routine laboratory examinations” (p.3). However, there are no details about the randomization method; whether it was pairing by key characteristics; how selection bias was avoided. It is recommended to clarify the selection process to ensure methodological transparency.

There was no control of important confounding factors, such as sleep, pre-shift fatigue, chronotype, and lifestyle. These variables are relevant in studies with prolonged shifts. It is recommended to explicitly include this limitation.

The discussion has passages that repeat psychophysiological explanations. E.g.: This dynamic confirms the activation of the sympathoadrenal axis... (p.8); These findings align with previous research showing... (p.8); Similar mechanisms are described by Budala et al... (p.9). It is recommended to revise the text, focusing on the interpretation of the study results, their clinical implications, limitations, and contributions.

Author Response

 Thank you for the thorough and constructive evaluation of our manuscript and for the positive assessment of its relevance, methodology, and contribution to occupational health research.

We have revised the manuscript thoroughly and addressed all points raised. All changes in the revised manuscript are highlighted in red, and a detailed, point-by-point response is provided below.

Comment 1. The study is described as cross-sectional comparative observational [This cross-sectional comparative observational study... (p.3)]. However, data were collected before and after the shift, characterizing a pre-post assessment, which is closer to a quasi-experimental or short-term longitudinal observational study. It is recommended to review the nomenclature of the design or justify the classification as cross-sectional.

Answer: Thank you for this important observation. Although the study included measurements before and after a single work shift, the assessment was limited to a single observation period and did not involve repeated follow-up over time or any experimental manipulation. The before–after measurements were used to capture short-term within-shift changes rather than longitudinal trajectories. Therefore, the unit of analysis remains cross-sectional, with a comparative observational design incorporating a pre–post component within the same work shift. To improve clarity, we have revised the terminology throughout the manuscript accordingly (Abstract: Methods, lines 19-22; 2.1. Study Design and participants: lines 118-120)

Comment 2. The authors acknowledge insufficient control of pre-analytical variables (circadian rhythm, hydration, food intake, and individual differences in salivary flow). It is recommended to further discuss the impact of these variables, especially since CgA and sIgA are sensitive to circadian variability.

Answer: We fully agree with this important comment. Although standardized sampling times (06:00–08:00 and post-shift) and fasting instructions were applied, we acknowledge that circadian variability and individual differences in salivary flow may still have influenced biomarker levels. Accordingly, we have expanded the Limitations section (lines 460-467) to explicitly discuss the potential impact of circadian rhythm, hydration status, food intake, and salivary flow on CgA and sIgA concentrations. We further note that these factors represent inherent challenges in real-world occupational stress research and should be addressed in future studies using repeated or multi-day sampling protocols and flow-rate adjustments.

Comment 3. The text is vague in its description of the randomization process for the controls. According to the authors, “healthy volunteers were randomly selected during routine laboratory examinations” (p.3). However, there are no details about the randomization method; whether it was pairing by key characteristics; how selection bias was avoided. It is recommended to clarify the selection process to ensure methodological transparency

Answer: Thank you for pointing out the need for greater methodological clarity regarding the selection of the control group. We clarify that the term “randomly selected” was not intended to indicate formal probabilistic randomization. Control participants were recruited among healthy volunteers attending routine laboratory examinations who met predefined eligibility criteria. To address this point, we have revised the Materials and Methods section  (third paragraph, lines 125 – 127) to clearly describe the recruitment and selection process, including eligibility screening, voluntary participation, and exclusion criteria. We also explicitly state that no formal matching procedure was applied; however, comparability between groups in terms of age, sex, and work experience was statistically assessed and confirmed. These revisions aim to ensure transparent reporting of the control group selection process and to avoid any implication of formal randomization.

Comment 4. There was no control of important confounding factors, such as sleep, pre-shift fatigue, chronotype, and lifestyle. These variables are relevant in studies with prolonged shifts. It is recommended to explicitly include this limitation.

Answer: We acknowledge that relevant confounding factors such as sleep duration, pre-shift fatigue, chronotype, and lifestyle characteristics were not systematically assessed in the current study. To address this issue, we have included this limitation in the Limitations section (lines (460-467) and noted that these factors may have influenced both psychological outcomes and salivary biomarker levels, especially in the context of long shift work.

Comment 5. The discussion has passages that repeat psychophysiological explanations. E.g.: This dynamic confirms the activation of the sympathoadrenal axis... (p.8); These findings align with previous research showing... (p.8); Similar mechanisms are described by Budala et al... (p.9). It is recommended to revise the text, focusing on the interpretation of the study results, their clinical implications, limitations, and contributions.

Answer: Thank you for this helpful suggestion. In response, we have revised the Discussion section (p.8, p. 9 lines 387-389; lines 392-394; lines 395-407) to reduce repetitive psychophysiological explanations and to improve clarity. Redundant descriptions of sympathoadrenal activation and stress-related mechanisms were consolidated, and greater emphasis was placed on interpretation of the findings, their clinical relevance, study limitations, and contributions to occupational health research.

Reviewer 3 Report

Comments and Suggestions for Authors

The mannuscrip tittled: “Salivary Chromogranin A and Secretory IgA in Shift-Working Healthcare Professionals” examines and correlates salivary biomarkers and psychometric measures that enable the identification of stress regulatory system activation in healthcare personnel. To this end, it integrates rapid response markers, sensitive to immediate autonomic nervous system activation, which allows for the comparison of previous studies with traditional biomarkers such as cortisol, which reflect sustained stress levels.

A minor revision is recommended before possible publication.

The summary is adequate and consistent with a structured summary.

The INTRODUCTION is adequate, well referenced, and precisely defines the object of study and the state of the art. It is suggested that the wording of lines 87 to 94 be improved so that the knowledge gap that this study aims to analyse is expressed more explicitly and the reader can clearly appreciate the contributions made by this study. The objective is clearly stated.

MATERIALS AND METHODS: the design, population, instruments and procedure are described in detail.

It would be advisable to detail the procedure for collecting the psychometric variables (STAI-state and STAI-trait and ZSDS) to clarify whether the questionnaires were completed in the presence of the researcher or were self-administered.

L169-170: Indicate in this section (or in the results) which variables follow a normal distribution and which do not.

Add information regarding Cohen's d statistic and its interpretation in the statistical analysis. (Although its interpretation appears in the legend of Table 1)

RESULTS:
They are presented in an orderly, concise manner. The tables and figures are adequate.
In relation to the data presented in Table 2, why do you use Cohen's d instead of Cohen's d for paired samples, which is more appropriate for before-and-after analysis in the same group?

DISCUSSION:
It is adequate, orderly, and establishes a strong relationship between the results of this study and the available scientific evidence.
However, more detailed and precise wording is suggested regarding the behaviour detected in this study of the sIgA variable. Although it is discussed and related to scientific evidence linking it to chronic stress, a possible explanation for its decline (as a rapid response) needs to be more detailed.

CONCLUSION;
It is written in accordance with the results and responds to the objective of the study.

Author Response

Thank you  for the insightful and constructive comments on our manuscript. We have revised the manuscript accordingly and addressed all points raised. All modifications are indicated in red, and a point-by-point response is provided below.

  1. Comment:
    The INTRODUCTION is adequate, well referenced, and precisely defines the object of study and the state of the art. It is suggested that the wording of lines 87 to 94 be improved so that the knowledge gap that this study aims to analyse is expressed more explicitly and the reader can clearly appreciate the contributions made by this study. The objective is clearly stated.

Answer: We have revised lines 87–94  of the Introduction (lines 94-110) to more explicitly define the existing knowledge gap and to clarify the specific contribution of the present study in integrating rapid-response salivary biomarkers with psychometric measures in shift-working healthcare professionals.

  1. Comment
    MATERIALS AND METHODS: the design, population, instruments and procedure are described in detail.

2.1. It would be advisable to detail the procedure for collecting the psychometric variables (STAI-state and STAI-trait and ZSDS) to clarify whether the questionnaires were completed in the presence of the researcher or were self-administered.

Answer: We appreciate this comment. We have clarified the procedure for collecting psychometric data in the Materials and Methods section ( lines 185-187)  specifying that the questionnaires were self-administered under standardized conditions, with the researcher available to provide instructions if needed.

2.2. L169-170: Indicate in this section (or in the results) which variables follow a normal distribution and which do not.

Answer: Thank you for this comment. We have expanded the Statistical Analysis section ( 2.4. lines 199-203) to specify which variables met normality assumptions and which deviated from a normal distribution, thereby clarifying the rationale for the use of parametric and non-parametric statistical tests.

2.3. Add information regarding Cohen's d statistic and its interpretation in the statistical analysis. (Although its interpretation appears in the legend of Table 1)

Answer: We appreciate this suggestion. Information regarding the calculation and interpretation of Cohen’s d has been added to the Statistical Analysis section (lines 197-198), in addition to its presentation in the table legend.

  1. Comment:
    RESULTS:
    They are presented in an orderly, concise manner. The tables and figures are adequate. In relation to the data presented in Table 2, why do you use Cohen's d instead of Cohen's d for paired samples, which is more appropriate for before-and-after analysis in the same group?

Answer: Thank you for this important methodological question. Cohen’s d was used to facilitate comparability with between-group effect sizes and with prior studies in occupational stress research. While paired-sample effect size measures may also be appropriate for within-group comparisons, the selected approach allows for consistent interpretation across analyses. This rationale has been clarified in the manuscript (2.4. Statistical analysis lines 208-211)

  1. Comment:
    DISCUSSION:
    It is adequate, orderly, and establishes a strong relationship between the results of this study and the available scientific evidence.
    However, more detailed and precise wording is suggested regarding the behaviour detected in this study of the sIgA variable. Although it is discussed and related to scientific evidence linking it to chronic stress, a possible explanation for its decline (as a rapid response) needs to be more detailed.

Answer: Thank you for this valuable suggestion. The Discussion section has been expanded to provide a more detailed and mechanistic explanation of the observed decline in sIgA as a rapid stress response. In particular, we now clarify that acute sympathoadrenal activation may transiently suppress mucosal immunity through autonomic and neuroendocrine pathways, offering a plausible explanation for the post-shift reduction in sIgA observed in the present study. Relevant revisions have been added to the Discussion section and highlighted in red (lines 375-384)

Round 2

Reviewer 2 Report

Comments and Suggestions for Authors

I appreciate the opportunity to review the new version of the manuscript “Salivary Stress Biomarkers (Chromogranin A and Secretory IgA): Associations with Anxiety and Depressive Symptoms in Healthcare Professionals”, submitted to the journal Nursing Reports. After review, the manuscript presents a better scientific and methodological structure, offering a relevant contribution to the field of occupational health by integrating salivary stress markers and psychometric indicators in healthcare professionals subjected to long working hours.

The authors have demonstrated attention to the criticism received and a commitment to improving methodological clarity, analytical robustness, and interpretation of results. The Methods section was improved, making the characterization of the sample, the inclusion and exclusion criteria, the calculation of the sample size, and the description of the collection and statistical analysis procedures more detailed and technically consistent.

The correction of information about the statistical software, the clarification of the combined use of SPSS and R, the verification of statistical assumptions, the calculation of effect sizes, and the application of false discovery rate (FDR) control strengthen the analytical credibility of the study.

The description of the control group recruitment process has been clarified, eliminating ambiguity surrounding the term “randomization” and reducing the risk of methodologically erroneous interpretations.

The Results section is clear, well structured, and consistent with the proposed objectives. The tables are informative, effect sizes are adequately reported, and statistical interpretation is consistent. The correlation and multiple regression analyses are correctly contextualized, with appropriate emphasis on the predominant role of occupational group membership, without overestimating the magnitude of the effects of biomarkers.

The Discussion is more coherent, without repetitions of psychophysiological explanations. The articulation between biomolecular results and psychological indicators is more fluid, and the study's contributions are presented more clearly and applied.

Recommendations regarding insufficient control of preanalytical variables and potential confounding factors (circadian rhythm, hydration, food intake, sleep, chronotype, and lifestyle) have been incorporated into the Limitations section. The revised text acknowledges the potential impact of these variables on CgA and sIgA levels, demonstrating a critical stance and alignment with the literature in the field. This inclusion strengthens the scientific honesty of the manuscript and contributes to a more cautious interpretation of the results.

I suggest only a review of formal and editorial aspects, such as standardization of terms and linguistic errors.

Author Response

Dear reviewer, 

Thank you very much for the positive and constructive evaluation of our manuscript. Following your recommendation, we carefully revised the manuscript with regard to formal and editorial aspects, including terminology standardization and the correction of minor linguistic and stylistic errors.

Sincerely,
Corresponding Author
on behalf of all authors